# Monitoring the Opening of Rapid Palatal Expansion (RPE) in a 3D-Printed Skull Model Using Fiber Optic F–P Sensors

**DOI:** 10.3390/s23167168

**Published:** 2023-08-14

**Authors:** Zhen Zhao, Shijie Zhang, Faxiang Zhang, Zhenhui Duan, Yingying Wang

**Affiliations:** 1Institute of Stomatology, Shandong University, Jinan 250012, China; 2Department of Orthodontics, Qilu Hospital of Shandong University, Jinan 250012, China; 3Faculty of Computer Science and Technology, Qilu University of Technology (Shandong Academy of Sciences), Jinan 250014, China; 4Institute of Automation, Qilu University of Technology (Shandong Academy of Sciences), Jinan 250014, China

**Keywords:** optical fiber sensors, Fabry–Perot sensors, orthodontic biomechanics, rapid palatal expansion

## Abstract

We present a novel method for the online measurement of multi-point opening distances of midpalatal sutures during a rapid palatal expansion (RPE) using fiber optic Fabry–Perot (F–P) sensors. The sensor consists of an optical fiber with a cut flat end face and an optical reflector, which are implanted into the palatal base structure of an expander and is capable of measuring the precise distance between two optical reflective surfaces. As a demonstration, a 3D-printed skull model containing the maxilla and zygomaticomaxillary complex (ZMC) was produced and a miniscrew-assisted rapid palatal expander (MARPE) with two guide rods was used to generate the midpalatal suture expansion. The reflected spectrums of the sensors were used to dynamically extract cavity length information for full process monitoring of expansion. The dynamic opening of the midpalatal suture during the gradual activation of the expander was measured, and a displacement resolution of 2.5 μm was demonstrated. The angle of expansion was derived and the results suggested that the midpalatal suture was opened with a slight V-type expansion of 0.03 rad at the first loading and subsequently expanded in parallel. This finding might be useful for understanding the mechanical mechanisms that lead to different types of expansion. The use of a fiber optic sensor for mounting the rapid palatal expander facilitates biomechanical studies and experimental and clinical evaluation of the effects of RPE.

## 1. Introduction

In recent years, fiber optic sensors have become increasingly attractive for biomechanical research [1,2,3], such as the detection of skeletal strain [4], the monitoring of forces induced by tendons and ligaments [1], stresses in intervertebral discs and in meniscus [1,5], chest wall deformation [2], biomechanical analysis of fracture healing [2], wearable motion and respiratory monitoring [1,3], and many measurement applications in dental biomechanics [6,7,8,9], etc. In dental biomechanics research, fiber optic sensors have the advantages of being waterproof compared to electronic sensors, not afraid of being attacked by saliva, good biocompatibility, small sensor size, multiplexing capability, and immunity to electromagnetic interference (EMI), providing a promising monitoring tool for dental biomechanics research.

Maxillary transverse deficiency (MTD) is one of the most common malocclusions at all ages, and its main treatment is maxillary skeletal expansion (MSE), routinely using a rapid palatal expansion (RPE) appliance. Traditionally, the use of RPE and surgically assisted rapid palatal expansion (SARPE) in late adolescent and adult patients has been considered to have some disadvantages [10,11,12]. To increase skeletal expansion and reduce the side effects of tooth-borne RPE, various types of bone-borne RPE appliances have been developed. In recent years, miniscrew-assisted rapid palatal expansion (MARPE) has become a new treatment method for late adolescent and adult MTD patients due to its significant advantages [13,14,15,16,17]; however, its device design, implantation site, expansion efficacy, and complications are also controversial [18,19,20]. These appliances may produce different results based on their design and active protocol. The effective opening of the midpalatal suture is an important basis for evaluating the effectiveness of the expander. The existing research methods on the opening of the midpalatal suture due to the excitation of the expander mainly include finite element analysis [20,21,22,23] and CT imaging [10,17]. The finite element method struggles to reproduce the fine structure of the complex jaw structure, the zygomaticomaxillary complex (ZMC), and the palatal expander structure, and CT imaging is complicated and difficult to monitor the complete process of palatal suture opening dynamically in real-time. In addition, there are often some differences in the results of the two studies. Therefore, it is necessary to be able to measure the dynamic changes in the opening process.

The use of fiber optic sensors for biomechanical studies of MSE has been investigated in very recent years, e.g., strain measurement of the Hyrax appliance using fiber Bragg grating (FBG) sensors in a 3D-printed human maxillary model [24] and in vitro validation of FBG sensors for rapid palatal expander suture expansion strain assessment [25], etc. Due to their small size, implantability, and on-line measurement, these sensors are able to reproduce the dynamic processes of the monitored parameters in RPE, providing important process parameters for diagnosis and evaluation of treatment effectiveness. However, their applications in the field of RPE monitoring are very recent and their practicality for full-scale implementation has not yet been fully established. Although the indirect response to the opening of the midpalatal suture through the monitoring of stress and strain parameters has been verified, no real-time monitoring of the opening displacement of the midpalatal suture during MSE has been reported.

In this paper, we propose a new method to monitor the opening of the midpalatal suture during RPE by mounting a pair of fiber optic Fabry–Perot (F–P) sensors on the palatal bases of an RPE appliance. As a demonstration, we used a 3D-printed scanned skull model fitted with a MARPE appliance to form a model simulation system for bone expansion simulation. F–P sensors were mounted on this model simulation system and while monitoring the opening distance of the midpalatal suture, we wanted to extrapolate the opening angle of the midpalatal suture from the comparison of the two sensors monitoring data. This model simulation system is more capable of simulating the complex structure of the ZMC than finite element simulation, maintains the structural integrity of the MARPE appliance, and provides a new evaluation method for the biomechanical study of the maxillary skeletal expander with better economy and reproducibility than in vivo experiments. The advantages of miniaturization and good biocompatibility of the fiber optic sensor also offer the possibility of clinical monitoring applications.

## 2. Materials and Methods

### 2.1. Sensor Concepts and Principles

A fiber optic Fabry–Perot (F–P) sensor is a device that uses the variation of reflected light intensity caused by the interference of light waves between two parallel mirrors separated by a small gap to measure physical parameters, such as displacement, pressure, temperature, or strain. A schematic diagram of our proposed fiber optic F–P sensor mounted on a MARPE appliance is shown in Figure 1. The MARPE appliance used in this study was a helical arch expander (Type 9162-310E, Shinye, Hangzhou, China), which includes two palatal base structures for mounting miniscrews, a central folding helical screw for providing displacement for mechanical expansion, and two guide rods. Activation of the appliance consisted of turning the central folding helical screw one-sixth of a turn (hexagonal screw configuration), in which one turn resulted in a 0.8 mm expansion. The fiber optic F–P sensor consists of an optical fiber and an optical reflector mounted on two base anchoring structures. The optical fiber (G657.A1, YOFC, Wuhan, China) was stripped of its coating at the end, had an outer diameter of 125 μm, and had a cut flat end face with an optical reflectivity of approximately 0.04. For collimation, a quartz tube jacket with an outer diameter of 400 μm and an inner diameter of 200 μm is placed over the end of the fiber. The optical reflector is a custom-made quartz fiber with a diameter of 400 μm, polished on the end face and coated with reflective film, with an optical reflectivity of 99%. An epoxy resin adhesive (9005, LEAFTOP, Shenzhen, China) was used for bonding the F–P sensors on the palatal base structures. At the end of the quartz tube jacket, the quartz tube jacket is also bonded to the optical fiber via epoxy resin adhesive. The initial distance was about 0.3 mm between the two end surfaces of the sensor. The incident light in the optical fiber is reflected at the fiber end face and at the optical reflector, and forms an interference. The variation of the cavity length of the interference cavity reflects the variation of the opening distance of the palatal bases in the MARPE appliance.

The two reflections possess a certain phase relationship determined using the optical path difference (OPD), defined as OPD = 2nd, n is the refractive index of the dielectric (*n* = 1 for air), and d is the physical cavity length, which represents the monitored open distance of the midpalatal suture. The two beams are coupled back into the lead-in fiber and interfere. The interference spectrum is then obtained to calculate the OPD encoded and can be expressed as [26]:(1)$$I_{r}=\frac{R_{1}+q^{{}_{2}}R_{2}-2q\sqrt{R_{1}R_{2}}\cos\delta }{1+q^{{}_{2}}R_{1}R_{2}-2q\sqrt{R_{1}R_{2}}\cos\delta }I_{i},$$
(2)$$\delta =\frac{4\pi nd}{\lambda }+\varphi_{0},$$
where *R*_1_ and *R*_2_ are the reflectivities of the optical fiber end and the optical reflector, and *R*_1_ = 0.04 and *R*_2_ = 0.99 are in the calculation of this paper, *I_i_* is the incident intensity, *q* is the light coupling coefficient, *λ* is the wavelength of the incident light, and *φ*_0_ is the additional phase.

Neglecting *φ*_0_, the wavelength *λ_i_* of the *i*th interference peak in the obtained optical spectrum is directly related to the cavity length d using:(3)$$\frac{4\pi nd}{\lambda_{i}}=\left[2k-(i-1)+1\right]\pi ,$$
where *k* is the corresponding interference order number of the first peek and should be an integer. Therefore, *k* can be obtained from the first and the *i*th peek as:(4)$$k=\frac{(2i-3)\lambda_{i}+\lambda_{1}/2}{2(\lambda_{i}-\lambda_{1})}.$$

Using the obtained *k*, the cavity length can be obtained as:(5)$$d=\frac{1}{4}(2k+1)\lambda_{1}.$$

Multiple peaks in the spectrum can be used to calculate the fringe order using Equation (4), avoiding the influence of the fringe order uncertainty, and the cavity length monitoring a large dynamic range can be realized.

The effect of temperature variation on the cavity length measurement is mainly caused by the mismatch in elongation due to the inconsistent coefficient of thermal expansion of different materials. If the coefficient of thermal expansion of the expander material is *α*_1_ and the coefficient of thermal expansion of the fiber material is *α*_2_, the cavity length temperature sensitivity of the sensor, defined as the ratio of the change in cavity length to the change in temperature, can be expressed as:(6)$$\frac{\Delta d}{\Delta T}=l\alpha_{1}-(l-d)\alpha_{2},$$
where *l* is the fixed length of the sensor. Using a fixed length of *l* = 6 mm, the thermal expansion coefficient of the expander material *α*_1_ = 15 × 10^−6^/°C and the thermal expansion coefficient of the optical fiber *α*_2_ = 5.5 × 10^−7^/°C, the cavity length temperature sensitivity of the sensor can be estimated to be 0.087 μm/°C. It can be seen that the sensor has a very low-temperature sensitivity and the temperature effect in the experiment can be neglected. The sensor is, therefore, also suitable for applications in environments where there is some variation in temperature.

### 2.2. Preliminary Test

In order to evaluate the capability of the sensors to measure displacement, preliminary tests were carried out, which were achieved by generating displacement through a precision translation stage and verifying the agreement of the measurements with the configured displacement. The schematic diagram of the experimental setup is shown in Figure 2a. A fiber optic end and an optical reflector are mounted on two separate translation stages and the central axes are aligned to ensure that the two end faces are parallel. The fiber end is connected to the optical interrogator for testing. During the test, the cavity length was gradually changed from the initial distance at equal intervals of 0.2 mm by adjusting the translation table, and the reflection spectrum was recorded at each step as shown in Figure 2b. The peaks in the spectrum were detected by the fiber interrogator and the measured displacements were derived from Equations (4) and (5). Figure 2c gives a linear relationship between the change in measured displacement and the configured displacement of the translation stage. As can be seen, there is an excellent agreement, demonstrating that the sensor can perform displacement measurements with high accuracy.

### 2.3. Simulation System Based on 3D-Printed Model

A human skull model, including ZMC, maxilla and teeth, was 3D-printed from a digitized scan of a female patient in late adolescence. The patient was informed about this study and the study was ethically compliant. An 8200 type photosensitive resin (WeNext Technology Co., Ltd., Shenzhen, China) with a tensile modulus of 2.6 Gpa was used to produce the 3D model. The modulus of elasticity of the model material is between that of cortical bone and cancellous bone (13.7 Gpa and 1.37 Gpa for cortical bone and cancellous bone, respectively [20,21]). We reproduced the maxillary midline separation by cutting a suture approximately 1 mm wide along the midline with a medical hand drill. Four miniscrews (diameter: 1.6 mm, length: 11 mm, Cibei, Ningbo, China) were used to anchor the arch expander. The MARPE appliance was anchored by the mini-screws in the maxillary mesial suture at a distance of 22 mm from the upper anterior molar, with its center aligned with the midpalatal suture line, as shown in Figure 3. The transverse mesial axis of the expander is located between the first premolar and the second premolar.

The MARPE appliance was opened with 3 activations, 1/6th of a turn each, by turning the hexagonal screw with a special wrench. This is generally considered to be the maximum amount of activation allowed to be applied in one operation. In clinical practice, a greater expansion rate would make the patient experience more discomfort, and increase the risk of causing side effects or failure. The opening distance of the two sensor positions was monitored in real-time during the operation. After each activation, the next operation was carried out after about 1 min in order to observe the change in the opening distance. A total of 1/2 turn was made and the opening distance was expected to be 0.4 mm according to the operating manual and experiences of operation.

The reflected spectrums of the F–P sensors were acquired using an optical interrogator (TV130, Tongwei Sensing, Beijing, China), with a wavelength range of 1510~1590 nm, wavelength repeatability of 2 pm, and an acquisition rate of 1 Hz, as shown in Figure 4. The interrogator has 4 channels for each synchronous monitoring, of which 2 channels are used to monitor the two F–P sensors. The peaks of the F–P sensor reflected spectrum was monitored using the interrogator software. The first and the last peaks were used to calculate the cavity length of the sensors, according to Equations (4) and (5), which reflect the opening distances of the midpalatal suture at the location of the sensor. Very high measurement accuracy can be achieved due to the detection of distance changes by interferometric spectroscopy. The measurement error mainly depends on the fitting accuracy of the interrogator to the spectral peak wavelength.

## 3. Results and Discussion

The reflected spectrums of the F–P sensors were observed when the expender with the sensors was mounted before the operation, as shown in Figure 5. The free spectral range (FSR), which is the distance between two adjacent wavelengths, was 3.9 nm for sensor1 and 3.85 nm for sensor2 at 1550 nm. It can be estimated that more than 20 peaks can be obtained in the spectral range of 80 nm, which ensures that there is a sufficient number of peaks to perform the operation of Equation (4). The initial distances of the two sensors can be calculated as 285 μm and 300 μm, respectively, according to Equation (5). As the cavity length increases, the FSR decreases and, thus, the number of peaks increases, still ensuring a sufficient number of peaks to perform the calculation of the cavity length.

The time response of the distances measured by the two sensors during the activations of the expander mounted in the skull model is shown in Figure 6. At first, the initial distance was acquired without operation. The measurement results fluctuated by ±2.5 μm (standard deviation value) when the expander was not activated, indicating that the sensor has a very high distance resolution.

According to the dynamic changes in the opening distance of the midpalatal suture by the two sensors in Figure 6, the amount of change from the initial distance monitored by each sensor (denoted by S1 and S2) separately for each loading, the average value (denoted by average) of the change for the two sensors were calculated and compared with the expected value (denoted by expected) based on experience, as shown in Figure 7. Where, based on experience, the expansion screw of the expander is considered to be activated for 1/6 turns each time, resulting in an expansion distance of 0.8/6 ≈ 0.13 mm. The results show that the average tested distance of the two sensors increases with the number of expansions, which agrees with the expected value with an error of 4 μm ± 7.35 μm (mean ± SD). This indicates that the expansion simulation system has a very high loading efficiency and that the sensor monitoring data are reliable and effective.

The fluctuation of the monitoring value during and after the opening operations in Figure 6 was ±9.6 μm, indicating the high monitoring accuracy also in the dynamic process of loading. The dynamic details of the loading process show that distance change is conducted at once by each loading and the distance remains at a stable value without degradation. This result shows that the expander has good loadability and retention of the expansion amount.

At the same time, the skeletal expansion pattern can be quantitatively described by comparing the monitoring data from the two sensors. Based on the results in Figure 6, for the first loading, the difference in the distance between the opening of the midpalatal suture monitored by the two sensors increased significantly, which suggests that a slight type V expansion of the midpalatal suture has occurred. Sensor 2 monitored a greater distance of opening, indicating that the midpalatal suture opening was wider in the anterior region than in the posterior region. The angle of expansion of the midpalatal suture can be deduced as 27/900 = 0.03 rad (1.72°) after the first loading, 0.021 rad (1.21°) after the second loading, and 0.031 rad (1.78°) after the third loading (where the distance of the two sensors is 900 μm). The above analysis is based on the assumption of left–right symmetric expansion. For comparison, we calculated the angle of expansion using data reported in the previously published literature on MARPE [17,27,28,29,30,31], as shown in Table 1.

Previous studies have reported differences in the amount of expansion between the anterior and posterior regions after MARPE. Although the results have varied, the majority of studies suggest that expansion in the anterior region is greater than that in the posterior region. The clinical results of the opening angle in [17] are larger than the results of this study. Opening angles reported in [27,28,29,30] are smaller and vary from 0.19° to 0.8°. Although the opening angle is very small and close to parallel expansion, it still conforms to the trend of the anterior region being larger than the posterior region. The finite element simulation in [31] also suggested a larger opening angle. The differences were generally attributed to the different relative positions of the center of resistance and the point of application of the force generated by the expander, as reported in [31]. However, this does not seem to adequately explain why the guide rods of the expander did not function as expected. With the guide rods, the expander is expected to expand in a parallel pattern.

One interesting measurement result is that with the increase in loading times, the difference in the opening distance of the midpalatal suture monitored by the two sensors did not increase further (shown in Figure 8a), suggesting that the opening angle did not increase. This may be related to the restraining effect of the guide rod of the expander, which has not been discussed in the literature. A reasonable explanation for this process is that during the initial loading, due to structural errors in the expander assembly (which is also corroborated by a photograph of the expander after expansion activation, as shown in Figure 8b), the guide bar did not fully function, which led to the onset of tilt with the overall mechanical properties of the jaw structure. During subsequent loading, the gradual expansion of the tilt was limited as the restraining effect of the guide rods took effect. The estimated opening angle at 1/2 turns of loading remained at approximately 0.031 rad (1.78°), indicating that the mechanically driven action leading to non-parallelism only occurred during the initial activation. This result suggests that in clinical practice, attention should be paid to each initial process during the expansion cycle of the arch expander. The operation of minimizing the proportion of the initial loading process in clinical practice may help to obtain a more parallel expansion. It also suggests that the guide rod should be modelled as a constraint when performing expander analysis and simulation with the finite-element method, that the structural and assembly errors of the expander should not be ignored when performing small loading, and that the structural design of the expander guide rods should also be considered to reduce structural errors at smaller expansions.

The effect of temperature variation on the cavity length measurement was tested by placing the sensor and the MARPE appliance in an oven, as shown in Figure 9a. Changing the temperature in the oven from 25 °C to 45 °C, the cavity length versus temperature variation curve of sensor2 was obtained as shown in Figure 9b. Through fitting, the temperature sensitivity of the cavity length was obtained as 0.1081 μm/°C, which is consistent with the theoretically calculated value of 0.087 in the previous paper. This suggests that the temperature sensitivity of the sensor is very low and is little affected by the ambient temperature.

Fiber optic sensor arrays can be used to measure the displacement of the appliance at multiple points simultaneously. In addition, it can be accompanied by strain, temperature, and other multiparameter measurements. Smaller appliances can be used by patients in clinical studies to help understand the biomechanical process of maxillary expansion. A biocompatible adhesive can be used to connect the FBG sensor to the application. The inherent safety of using light rather than current, as well as the biocompatibility of the fiber optic, can enable the application of the current appliance to be considered in more in vitro or clinical studies.

## 4. Conclusions

A methodology based on fiber optic F–P sensors to measure the multi-point opening of the expander during maxillary expansion is proposed to dynamically monitor the expansion process of a MARPE with guide rods on a 3D-printed maxillary-based model. The opening distance measurements of these sensors were in agreement with the operating manual and experiences of operation. The sensor was demonstrated to have a resolution of 2.5 μm and to be capable of monitoring the angle of expansion. A difference in the amount of opening between the two monitoring points at the first loading was monitored, suggesting the occurrence of a type V expansion, with an angle of 0.03 rad. The angle did not expand further with the subsequent expansion. This reveals that the mechanical action that leads to the expansion angle occurs at the initial loading. This provides a means to study the dynamic expansion process of the palatal expansion apparatus. In vitro testing is expected to contribute to the development of such mechanical structures and clinical procedures, such as the ability to better understand the process of RME expansion and to better control the amount of expansion and the angle of dilation required for RME treatment. Because of the advantages of miniaturization, high sensitivity, and good biocompatibility of fiber optic sensors, future research includes the study of dynamic dilation processes in more types of arch expanders and the evaluation of possibilities for clinical applications.

## Figures and Tables

**Figure 1 sensors-23-07168-f001:**
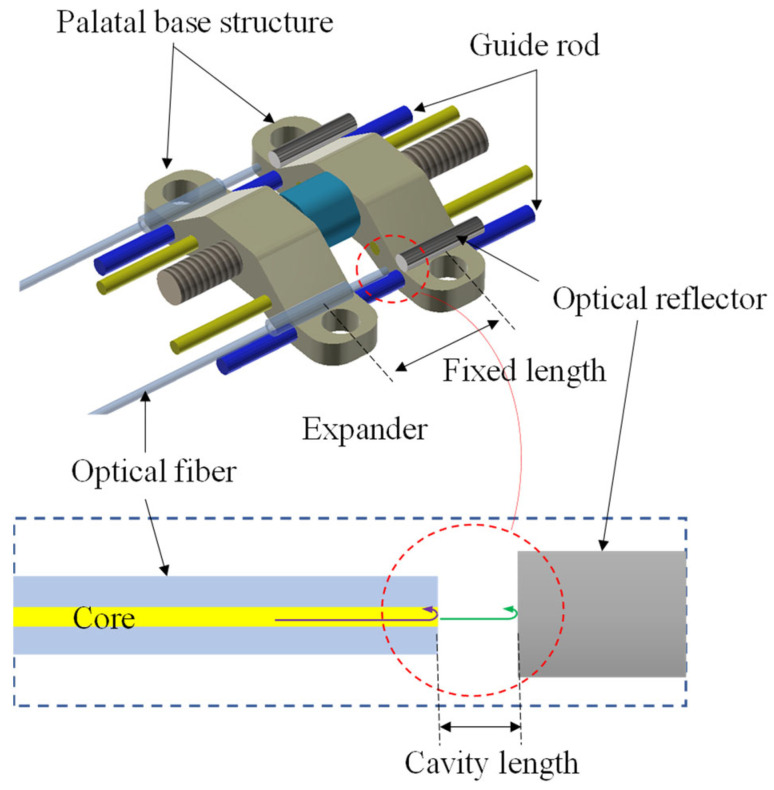
Schematic diagram of a fiber optic F–P sensor mounted on a MARPE appliance with two guide rods, where the inset is the optical schematic of the F–P sensor.

**Figure 2 sensors-23-07168-f002:**
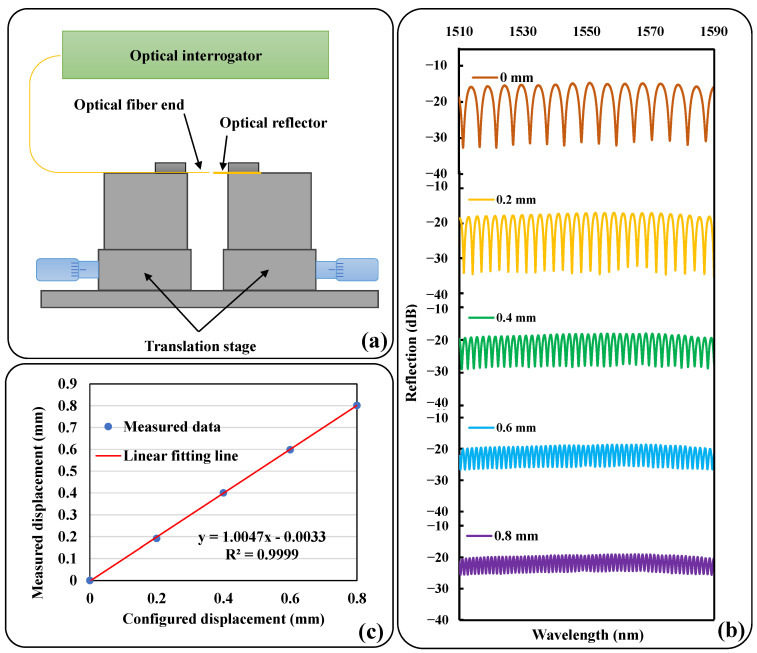
(**a**) The distance tuning device for preliminary test; (**b**) reflection spectra recorded at each step; and (**c**) the measured displacement with respect to the configured displacement. The straight red line is a linear fitting line.

**Figure 3 sensors-23-07168-f003:**
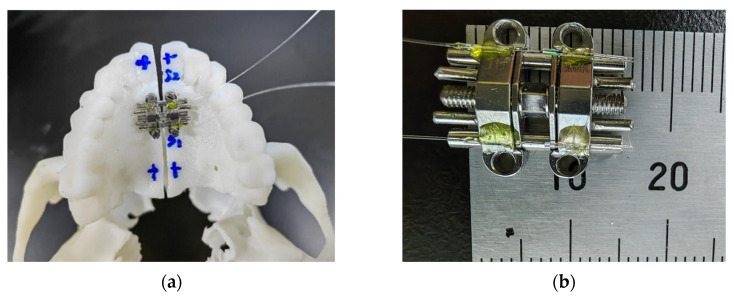
(**a**) Top view of the F–P sensor and the expender positioned in the 3D-printed model. (**b**) Detailed view of the F–P sensors mounted in the expander.

**Figure 4 sensors-23-07168-f004:**
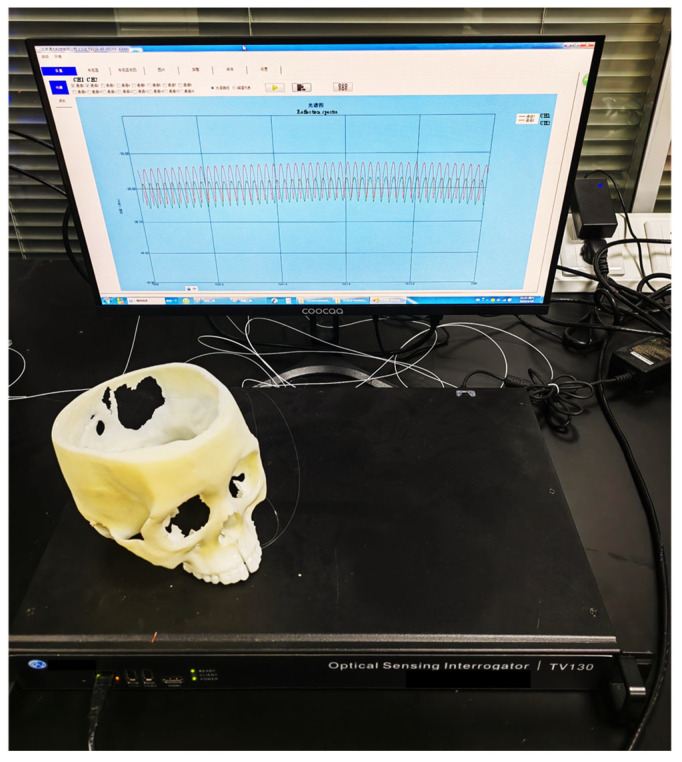
Experimental setup with F–P sensor connected to the optical interrogator TV130.

**Figure 5 sensors-23-07168-f005:**
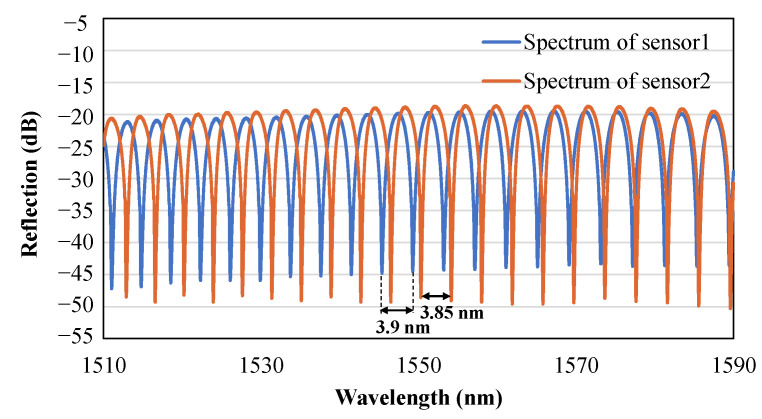
The reflected spectrum of the F–P sensors with an analysis of the FSR for each sensor.

**Figure 6 sensors-23-07168-f006:**
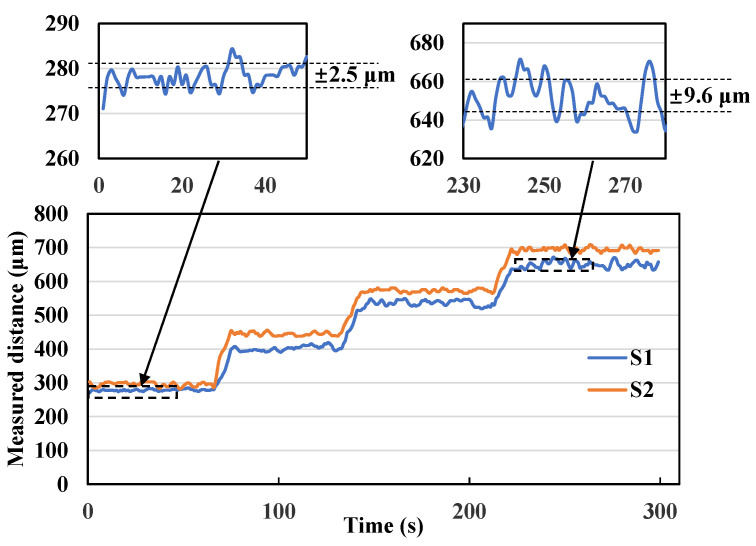
Dynamic changes in the opening distance of the midpalatal suture monitored by two sensors, in which fluctuations in monitoring results before and after loading are amplified.

**Figure 7 sensors-23-07168-f007:**
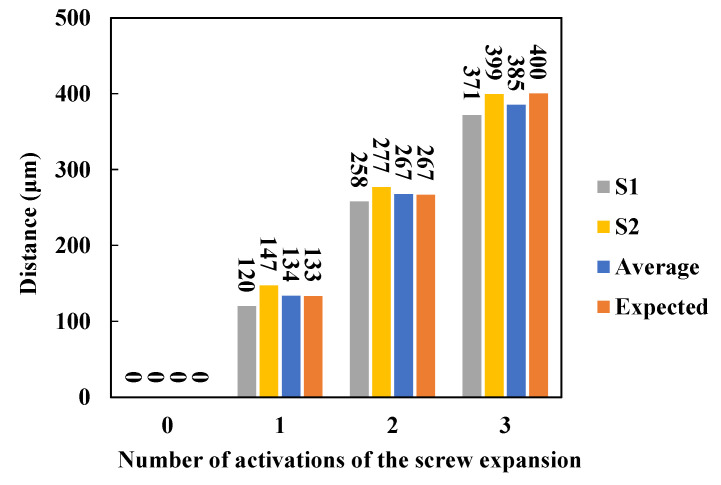
For each loading, the amount of change in the open distance of the midpalatal suture monitored by the sensors (S1, S2); the average value (Average) of the change monitored by the two sensors compared with the expected value based on experience (Expected).

**Figure 8 sensors-23-07168-f008:**
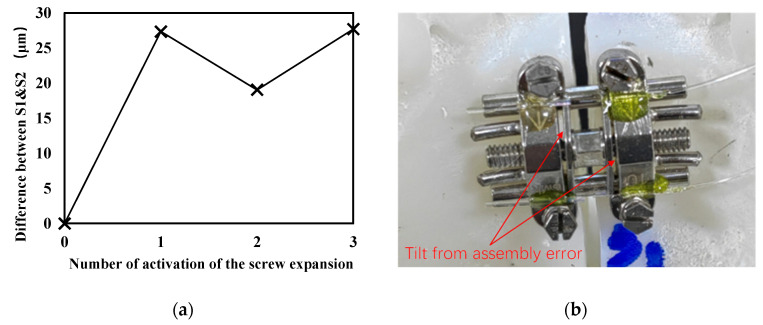
(**a**) Change in the difference between the two sensors (S1, S2) monitored at the opening of the midpalatal suture under each load (the initial difference is considered to be 0) and (**b**) photograph of the expander after expansion activation.

**Figure 9 sensors-23-07168-f009:**
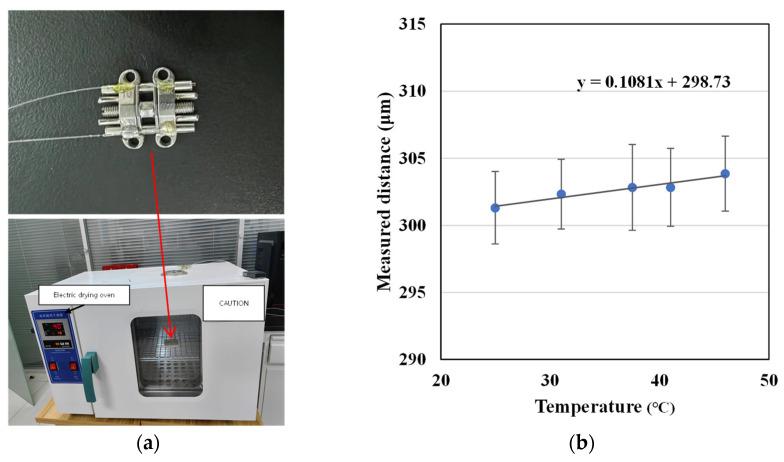
(**a**) Temperature response test of the sensor and (**b**) the test results.

**Table 1 sensors-23-07168-t001:** Comparison of different studies on the angle of expansion.

Study	Methodology	Expansion Pattern	Angle of Opening
Jia et al., 2021 [17]	CBCT *	Type V	2.89° **
Song et al., 2019 [27]	CBCT *	Parallel	0.19° **
Elkenawy et al., 2020 [28]	CBCT *	Parallel	0.24° **
Cho et al., 2022 [29]	CBCT *	Parallel	0.60° **
Liao et al., 2022 [30]	CBCT *	Parallel	0.8° **
Lee, et al., 2017 [21]	Finite element analysis	Type V	3.2° **
Chang et al., 2023 [31]	CBCT * and Finite element analysis	Parallel	0.57° **
This study	3D-printed model and sensors	Type V	1.78°

* Cone-beam computed tomography (CBCT) scans. ** Distance between anterior nasal spine (ANS) points and posterior nasal spine (PNS) points was assumed to be 50 mm.

## Data Availability

Data from the laboratory tests carried out in this study are available upon request to the corresponding author.

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
