# Peer review of "Monitoring the Opening of Rapid Palatal Expansion (RPE) in a 3D-Printed Skull Model Using Fiber Optic F–P Sensors"

_sensors, 2023, doi:10.3390/s23167168_

Round 1

Author Response

Thank you very much for giving us an opportunity to revise our manuscript, we have studied reviewers’ comments carefully and have made revision which marked in red in the paper. The point-by-point responses to reviewer's comments are in the response letter.

Reviewer 2 Report

The authors online measurement of pening distance of midpalatal suture during RPE using fiber FP sensors, which can be useful for clinic application. This method is simple and effective, it was clearly expressed. However, there are some issues need to be revised before being accepted.

1. Outer diameter of fiber is 125 um, but quartz tube has inner diameter of 200 um, will the gap influence?

2. Actual parematers, reflectivity of reflector is 0.99, cavity lenght is 0.3 mm. However, they are 0.04 and 0.6mm in the theoritical part.

3. FSR reading in Fig. 5 are 3.8 nm and 3.9 nm while at differ initial wavelength locations (FSR are wavelength dependened). You should double check whether your calculation are correct or not.

4. What is the error calculated from the average data and expected data in Fig 7?

5. 27/900=0.03 rad in the text, is 900 the distance of this two sensors? authors should indicate this.

good

Author Response

(The authors gave the same response as above.)

Round 2

Reviewer 1 Report

This version could be accepted.